# Characterization of HER2-Low Breast Tumors among a Cohort of Colombian Women

**DOI:** 10.3390/cancers16183141

**Published:** 2024-09-12

**Authors:** Laura Rey-Vargas, Lina María Bejarano-Rivera, Diego Felipe Ballen, Silvia J. Serrano-Gómez

**Affiliations:** 1Cancer Biology Research Group, National Cancer Institute, Bogotá 111411, Colombia; lrey@cancer.gov.co (L.R.-V.); lbejarano@cancer.gov.co (L.M.B.-R.); 2Doctoral Program in Biological Sciences, Pontificia Universidad Javeriana, Bogotá 110231, Colombia; 3Clinical Oncology Unit, National Cancer Institute, Bogotá 111411, Colombia; dballen@cancer.gov.co; 4Research Support and Follow-Up Group, National Cancer Institute, Bogotá 111411, Colombia

**Keywords:** HER2-low, antibody–drug conjugate, breast cancer, clinical pathological features, survival

## Abstract

**Simple Summary:**

HER2-low breast cancer is a newly recognized subtype that has shown promising responses to treatment with antibody–drug conjugates (ADCs). However, there is still little known about its clinical and molecular characteristics. In this study, we examined the clinical and pathological features, survival rates, and expression of HER2-related genes in Colombian patients with HER2-low breast cancer, comparing them to HER2-negative and positive groups. We found that HER2-low tumors were better differentiated and had a lower proliferation index compared to HER2-positive tumors. Additionally, compared to HER2-negative cases, HER2-low patients had higher mRNA expression of the *ERBB2* gene and longer overall survival rates. Despite these findings, there were no significant differences in survival when adjusted for estrogen receptor status and clinical stage. These results highlight the need for further research on HER2-low breast cancer to optimize treatment strategies for this unique group.

**Abstract:**

HER2-low tumors have shown promise in response to antibody–drug conjugates (ADCs) in recent clinical trials, underscoring the need to characterize this group’s clinical phenotype. In this study, we aimed to explore the clinicopathological features, survival rates, and HER2 amplicon mRNA expression of women affected with HER2-low breast cancer, compared with HER2-negative and HER2-positive groups. We included 516 breast cancer patients from Colombia, for whom we compared clinicopathological features, mRNA expression of three HER2 amplicon genes (*ERBB2, GRB7* and *MIEN1*), survival and risk of mortality between HER2-low cases (1+ or 2+ with negative in situ hybridization (ISH) result) with HER2-positive (3+ or 2+ with positive ISH test) and HER2-negative (0+) cases. A higher proportion of patients with better-differentiated tumors and a lower proliferation index were observed for HER2-low tumors compared to the HER2-positive group. Additionally, HER2-low tumors showed higher mRNA expression of the *ERBB2* gene and longer overall survival rates compared to HER2-negative cases. Nonetheless, a Cox-adjusted model by ER status and clinical stage showed no statistically significant differences between these groups. Our results show differences in important clinicopathological features between HER2-low and both HER2-positive and negative tumors. Given this unique phenotype, it is crucial to evaluate the potential advantages of ADC therapies for this emerging subtype of breast cancer.

## 1. Introduction

Human epidermal growth factor receptor 2 (HER2), a widely known biomarker for breast cancer prognosis assessment, has become one of the most important tools for oncologists to decide a patient’s treatment, as monoclonal antibody-based therapies designed to block HER2 activity, such as trastuzumab or pertuzumab, and tyrosine kinase inhibitors (TKIs), like lapatinib, have shown to dramatically reduce disease burden and improve efficacy outcomes in different breast cancer settings [1,2,3,4,5]. Several clinical trials have consistently demonstrated that patients with breast cancer benefit from targeted therapies, such as HER2-targeted agents, only if they exhibit HER2 protein overexpression or amplification of the *ERBB2* gene confirmed via in situ hybridization (ISH) techniques [6,7]. The current indication for HER2 assessment is based on the American Society of Clinical Oncology/College of American Pathologists (ASCO/CAP) guidelines, where HER2 positivity is defined by a complete and intense membrane immunohistochemistry (IHC) staining in ≥10% of cells (score 3+) and/or *ERBB2* amplification HER2/CEP17 ratio of ≥2.0 [8].

Among HER2-negative tumors, the landscape is much more complex. Tumors that do not show any degree of membrane staining are classified as 0+, whereas tumors that either present incomplete staining in >10% of tumor cells (score 1+) or a weak-to-moderate complete membrane staining in >10% of tumor cells (score 2+) but with a negative ISH result, are given a higher IHC score but are still defined as HER2-negative cases [9]. Studies that have compared *ERBB2* gene expression between tumors with either dim or incomplete IHC HER2 staining and negative HER2 expression have reported higher mRNA levels of *ERBB2*, validating the higher presence of this tyrosine kinase receptor within these tumor cells [10,11]. The dichotomous definition has been rethought over the past few years as a result of new emerging therapies that seem to benefit patients whose tumors express low levels of the HER2 protein (either 1+ or 2+ scores) [12].

Clinical trials have now been reporting significantly longer progression-free and overall survival rates in metastatic breast cancer patients with HER2-low tumors treated with antibody–drug conjugates (ADCs), such as trastuzumab deruxtecan (T-DXd), compared to the physician’s choice of chemotherapy [12,13]. The clinical success of these new therapies is attributed to the combined effect of the HER2 inhibitor trastuzumab linked to a cytotoxic agent, which, once released in the cell, can diffuse into the adjacent neighboring cells and induce non-antigen dependent cytotoxicity, having a broader effect on both HER2-overexpressing and non-overexpressing tumor cells [12]. Acknowledging these promising therapies, the latest ASCO/CAP guidelines for HER2 testing and evaluation include best practice recommendations to distinguish subtle differences between tumors with 0+ and 1+ scores [8].

Studies conducted on Caucasian and Asian women have reported high HER2-low prevalence rates among negative cases, with values that range between 31% and 60% [14,15,16,17]. This indicates that HER2-low tumors constitute a significant proportion of all breast cancer cases classified as HER2-negative, underscoring the importance of paying special attention to this potentially beneficial group of patients. It is well known that the distribution of breast cancer intrinsic subtypes (i.e., luminal, HER2-enriched, and triple-negative) varies considerably between Latinas and other population groups [18,19,20,21], suggesting that HER2-low prevalence among Latinas might also differ considerably from what has been reported in other cohorts. This hypothesis is supported by previous studies conducted on Latina women from Colombia, where higher mRNA levels of *ERBB2*, the gene encoding the HER2 receptor, have been found in women with greater contributions of Indigenous-American (IA) ancestry [22], which is a significant genetic component among the Latin-American population [23]. This pattern was also observed for other HER2 amplicon genes, such as *GRB7* and *MIEN1* [22]. Interestingly, this association was independent of the genomic amplification of HER2 [22], suggesting that Latina women with a significant contribution of IA ancestry will exhibit higher *ERBB2* expression. However, this expression may not be sufficient to generate complete HER2 membrane staining when evaluated by IHC, resulting in incomplete membrane staining and a higher prevalence of the HER2-low subtype among Latinas. Taking this into account, it is important to study and characterize the HER2-low subtype in a highly admixed population like Colombia.

New emerging evidence has led to the proposal that HER2-low tumors might represent a new subgroup of breast cancer cases in terms of prognosis, with its own response to treatment and therapeutic options [24,25]. In order to test this hypothesis, in this study, we aimed to explore the clinicopathological features, survival rates, and HER2 amplicon mRNA expression of Colombian women affected with HER2-low breast cancer compared to HER2-negative (0+) and HER2-positive groups (2+ with ISH+ or 3+), with the expectation that it will lead to a better understanding of this disease clinical features in the Colombian population.

## 2. Materials and Methods

### 2.1. Sample Selection

We included 516 breast cancer patients diagnosed between 2013 and 2015 at different health institutions in the country (Colombian National Cancer Institute (NCI) in Bogotá D.C. = 361, San Pedro Hospital (SPH) in Pasto = 54, Fundación Valle de Lili (FVL) University Hospital in Cali = 73, and Las Américas Clinic (LAC) in Medellin = 28). Eligibility criteria included histologically confirmed diagnosis of Invasive Ductal Carcinoma of No Special Type (NST), according to the latest World Health Organization criteria [26], and availability of formalin-fixed paraffin-embedded (FFPE) tissue blocks with at least 10% of tumor content from mastectomies or breast-conserving surgeries (quadrantectomies). This research was approved by the Colombian NCI ethics committee and defined as risk-free; therefore, according to Colombian laws, it was considered that no informed consent was required.

### 2.2. Immunohistochemistry

IHC assays were performed on 3 µm thick sections from a single FFPE surgery block with the highest tumor content, using monoclonal antibodies for estrogen receptor (ER) (clone SP1 Roche 05278406001), progesterone receptor (PR) (clone 1E2 Roche 05278392001), HER2 (clone 4B5 Roche 05278368001), and Ki67 (clone 30-9 Roche 05278384001) using the Roche Benchmark XT automated slide preparation system (Roche Ltd., Basel, Switzerland). Positive controls were included and 3,3′ diaminobenzidine (DAB) was used as the chromogen.

A single pathologist analyzed the IHC expression of ER, PR, HER2 and Ki67. The status of hormone receptors was considered positive when they exceeded 1% of nuclear staining in tumor cells. HER2 evaluation followed the recommendations of the ASCO/CAP guidelines [8] and was defined as follows: positive (3+) for complete and intense circumferential membrane within >10% of tumor cells; ambiguous (2+) for incomplete and/or weak/moderate circumferential membrane staining within >10% of tumor cells, or complete membrane staining but within ≤10% of tumor cells; negative (1+) for incomplete faint membrane staining within >10% of tumor cells; and negative (0+) for absence of staining. Based on ER, PR, HER2, and Ki67 IHC expression, we classified tumors into intrinsic subtypes according to the recommendations of the 2017 St. Gallen consensus [27].

### 2.3. Gene Expression Analysis

Before RNA extraction, a pathologist evaluated hematoxylin and eosin-stained slides to estimate the percentage of tumor representation in the paraffin block selected for each case. For FFPE blocks with 60% or higher tumor content, five sections of 5 μm were obtained, whereas for FFPE blocks with less than 60% of tumor content, 2–3 tumor cores were obtained using a 1 mm punch needle. RNA was extracted using the Qiagen isolation kit AllPrep DNA/RNA FFPE following the manufacturer’s protocol, and nucleic acid concentration was quantified using a NanoDrop ND1000 Spectrophotometer (Thermo Scientific, Wilmington, NC, USA).

We evaluated *ERBB2*, *GRB7*, and *MIEN1* gene expression using real-time RT-PCR. For reverse transcription, SuperScript™ III First-Strand Synthesis SuperMix (Invitrogen, Waltham, MA, USA) was used according to the manufacturer’s protocol, starting from an equal amount of 300 ng of RNA. *ERBB2* (Hs01001580_m1), *GRB7* (Hs00917999_g1) and *MIEN1* (Hs00260553_m1) TaqMan probes were used to quantify the levels of mRNA expression, using GAPDH (Hs03929097_g1) as the housekeeping gene. The reaction was amplified using the TaqMan^®^ Fast Advanced Master Mix (Applied Biosystems, Waltham, MA, USA) in a QuantStudio 3 Real-Time PCR instrument (Thermo Scientific, Waltham, MA, USA). Gene expression change analysis was performed using RNA from paired non-tumor paraffin blocks, and the 2^−ΔΔCT^ method was applied.

### 2.4. Statistical Analysis

Breast cancer patients were classified according to HER2 expression into the following groups: positive (3+ or 2+ with ISH+), negative (0+), and low (1+ or 2+ with ISH-). We applied a Chi-square or Fisher’s exact test to evaluate clinicopathological features for HER2-low cases compared with HER2-negative (low vs. negative) and HER2-positive tumors (low vs. positive). Additionally, a Wilcoxon Rank-Sum test was employed to examine differences in log fold change (FC) among HER2 amplicon genes *ERBB2*, *GRB7*, and *MIEN1* across the aforementioned groups. We evaluated differences in overall survival (OS) and recurrence-free survival (RFS) according to HER2 expression using the Kaplan–Meier and log rank test. OS was calculated from the date of diagnosis to the date of death or last follow-up. DFS was calculated from the date of surgery to the date of the first recurrence (local, regional, or distant relapse) or last follow-up. The risk of mortality was assessed using a Cox proportional hazard model adjusted for ER status and clinical stage. All statistical analyses were performed using the RStudio software, version 2024.04.1+748, and differences were considered statistically significant if *p* < 0.05.

## 3. Results

### 3.1. Patients’ Characteristics

Patients’ clinicopathological characteristics are described in Table 1. The patients were mostly diagnosed over the age of 50 (71.5%), in IIa/IIb clinical stages (43.4%), with moderately differentiated tumors (Scarff–Bloom–Richardson II: 51.8%) and sizes that ranged between 21 and 49 mm (37.0%). More than half of the patients had histological invasion (51%), lymph node involvement (54.5%), and presented high proliferation indexes (Ki67 ≥ 20%: 53.7%). In terms of disease management, the majority of patients received neoadjuvant therapy (54.4%), of which 36% underwent a cytotoxic regimen. Surgical management primarily involved mastectomies (50.7%). For adjuvant therapy, most patients received hormonal therapy (39.5%), followed by a combined regimen of cytotoxic and hormonal therapy (19.4%). Anti-HER2 therapy was received by only 12.6% of patients in the adjuvant setting. Additionally, most patients (82.2%) were treated with radiotherapy.

Regarding HER2 status, 325 cases (63%) were classified as negative (0+), 97 (18.8%) as low, (1+/2+), and 94 (18.2%) as positive (3+). ER+/HER2− (luminal A-like) (32.2%) and ER+/HER2− (luminal B-like) (32.2%) tumors were the most common intrinsic subtypes, followed by ER−/HER2− (13.4%), ER+/HER2+ (10.9%), and ER−/HER2+ tumors (7.4%). After 5 years of follow-up, 22.9%% of the patients presented clinical recurrence and 19.4% had died.

### 3.2. Clinicopathological Characteristics and HER2 Amplicon Gene Expression of HER2-Low Breast Cancer Patients

We evaluated the clinicopathological characteristics of HER2-low tumors and compared them to the other HER2-expressing groups (Table 2). This analysis showed more statistically significant differences between HER2-low and HER2-positive tumors than HER2-negative cases. A higher proportion of patients diagnosed over the age of 50 (79.4% vs. 60.6%, *p* = 0.008) with better-differentiated tumors (Bloom–Richardson II: 58.8 vs. 40.2%, *p* = 0.015) and lower proliferation index (Ki67 < 20%: 50.5% vs. 19.1%, *p* < 0.001) was observed among HER2-low tumors compared to the HER2-positive group. Additionally, as expected, patients with HER2-positive tumors were more frequently treated with the trastuzumab regimen during both neoadjuvant (55.8% vs. 12.7%, *p* < 0.001) and adjuvant therapy (57.7% vs. 10%, *p* < 0.001). When we compared these clinical features among HER2-low and negative tumors, a lower proportion of patients with disease progression after neoadjuvant therapy among the HER2-low group was observed (7.7% vs. 23.0%, *p* = 0.032). It is notable that when the type of neoadjuvant therapy between both groups was compared, there was a tendency, although not statistically significant, for a higher frequency of a cytotoxic plus trastuzumab regimen among HER2-low patients compared to the negative group (12.7% vs. 3.4%, *p* = 0.053).

On the other side, a higher proportion of ER (89.7% vs. 59.6%, *p* < 0.001) and PR-positive (81.4% vs. 47.9%, *p* < 0.001) cases was observed among HER2-low tumors compared to the positive group (Table 2). In concordance with these results, most HER2-low cases were classified as ER+/HER2− (luminal B-like) tumors (61.8%), followed by the ER+/HER2− (luminal A-like) subtype (34.2%). Only 3 out of 97 HER2-low tumors were classified as ER−/HER2− (3.9%), whereas among the HER2-negative group, 20.3% of cases were assigned to this subtype. Regarding HER2-positive cases, the majority were classified as ER+/HER2+ (59.6%). A further analysis was conducted to evaluate differences in clinicopathological variables between HER2 1+ and 2+ cases, but no statistically significant results were found (Appendix A).

Expression levels of the HER2-codifying gene, *ERBB2*, along with the other two biologically important genes located at the HER2 amplicon (17q12), *GRB7* and *MIEN1*, were assessed and compared between the analyzed groups (Figure 1). As expected, all three amplicon genes exhibited statistically significant overexpression in the HER2-positive tumors compared to the HER2-low group (log FC *ERBB2*: 3.41 vs. 0.75, *p =* 0.0023; *GRB7*: 0.85 vs. 0.21, *p* < 0.001; *MIEN1*: 2.23 vs. 0.83, *p* < 0.0075; respectively) (Figure 1a). Interestingly, when comparing HER2-low tumors to negative cases, a statistically higher expression was observed for the *ERBB2* gene (log FC 0.75 vs. 0.51, *p* = 0.011, respectively), while no significant differences were found for *GRB7* (*p* = 0.62) or *MIEN1* (*p* = 0.09) (Figure 1b). Furthermore, we also compared mRNA expression of the HER2 amplicon genes between HER2 1+ and 2+ cases and found no statistically significant differences (Appendix A).

### 3.3. Survival and Risk of Mortality of HER2-Low Breast Cancer Patients

Five-year survival (60 months) was assessed among HER2-low patients and compared to each of the HER2-expressing groups. Interestingly, this univariate analysis showed that HER2-low patients presented significantly longer OS median times compared to HER2-negative cases (low: 57.2 months vs. negative: 52.8 months, *p* = 0.025). In contrast, when compared to the HER2-positive group, no statistically significant differences in OS were observed (low: 57.2 months vs. positive: 54.7 months, *p* = 0.093) (Figure 2a,b). Similarly, median RFS times did not differ significantly between HER2-low and either of the HER2-expressing groups (low: 48.6 months vs. positive: 47.2 months, *p* = 0.75; low: 48.6 months vs. negative: 48.6 months, *p* = 0.82) (Figure 2c,d).

Based on these results, we evaluated the risk of mortality with a multivariate Cox model adjusted for ER status and clinical stage (Table 3). Contrary to our previous OS findings, the multivariate analysis revealed that breast cancer patients with HER2-low tumors do not experience better outcomes compared to those with HER2-positive (HR = 0.69, 95% CI, 0.30–1.61, *p* = 0.403) or HER2-negative tumors (HR = 0.57, 95% CI, 0.30–1.09, *p* = 0.092).

## 4. Discussion

The latest results from clinical trials like DESTINY-Breast04, involving ADC therapy for patients with metastatic breast cancer and confirmed HER2-low expression (1+ or 2+ with negative ISH), have substantiated the promising benefits of T-Dxd compared to standard chemotherapy [13,28]. Based on the bystander effect of these conjugated molecules, recent clinical trials have established an approximate threshold of at least ~100,000 expressing HER2 molecules in the cell surface (equivalent to a 1+ IHC score) for these therapies to confer a clinical benefit [13]. These findings provide an opportunity for a significant number of patients with low HER2 expression and limited treatment alternatives who, prior to this breakthrough, may not have had access to alternative therapeutic approaches. As a result, these novel molecules have drawn attention to these previously unexplored groups of patients, prompting a shift in the paradigm of conventional treatment approaches for breast cancer [17]. Following this era of clinical advances, we found it critical to characterize a cohort of breast cancer patients with HER2-low tumors from a national cancer reference center in Colombia, aiming to shed light on important clinical and pathological differences within this subset of patients when compared to both well-characterized HER2-negative and HER2-positive groups, hoping that our results will contribute to a more comprehensive understanding of the clinical features of this disease in the Colombian population.

Overall, our results suggest that patients with HER2-low breast tumors often exhibit clinical features associated with a better prognosis, such as a well-differentiated phenotype and a lower proliferation index; this was particularly evident when compared to the HER2-positive group. Additionally, a lower proportion of disease progression after neoadjuvant therapy was observed among HER2-low tumors when compared to the HER2-negative group, which might be related to the differences in the neoadjuvant regimens received in each group. In line with these findings, this subset of patients also showed longer OS times than HER2-negative cases; however, no association between HER2-low status and disease recurrence was observed. This suggests that a low-to-moderate expression of HER2, along with other associated clinicopathological features found in this group (well-differentiated tumors and a lower Ki67), may not exert a significant impact on preventing disease recurrence. However, it is possible to hypothesize that other confounding variables, like adjuvant treatment or additional factors related to disease management, might be playing a part in the outcome observed for breast cancer patients in this study.

Considering that hormone receptor (HR) expression among this set of tumors was significantly higher, we incorporated this parameter into a multivariate Cox model and confirmed that our previous findings of longer OS times in HER2-low patients compared to the negative group were probably driven by a higher proportion of ER-positive patients within the HER2-low group. Similar findings have been documented previously, indicating that HER2-low tumors not only exhibit a positive correlation with ER expression and a lower Ki67 index but also show associations with other clinically significant biomarkers such as the androgen receptor (AR), COX2 and pAKT [17]. While additional associations between HER2-low status and the expression of biomarkers and various clinicopathological features have been reported, the main consensus is that the characteristics of HER2-low cancers are predominantly influenced by HR status [17]. Therefore, it is imperative to consider HR status in the management of HER2-low breast tumors.

Consistent with the previous statement, earlier studies on the molecular characterization of HER2-low tumors have indicated that this subgroup frequently exhibits a mutational profile closely aligned with a luminal phenotype, supported by their elevated prevalence of *PIK3CA* and *GATA3* mutations, as well as a higher subclassification within the luminal A and luminal B subtypes according to PAM50 [29]; this aligns with our findings, where the luminal-like subtypes were also the predominant tumor classification among HER2-low tumors. It has been speculated that this particular phenotypic profile is related to a crosstalk mechanism between ER and HER2, where one receptor upregulates the other [30]. This aligns with findings indicating higher *ERBB2* mRNA levels in HR+/HER2-low tumors compared to the HR−/HER2-low group [14]. This bidirectional signaling is actually associated with the development of tumor resistance to endocrine or anti-HER2 therapies, as targeting either pathway leads to the upregulation of the other [30]. This explains why often a reduced pathological complete response (pCR) to neoadjuvant therapy is reported for patients with HER2-low tumors [31,32]. In our study, we found better responses to neoadjuvant treatment among HER2-low patients, although this was only observed when compared to the HER2-negative group, which is highly enriched with tumors with a triple-negative phenotype.

The mRNA expression analysis for the HER2 amplicon genes showed interesting results. As expected, all three assessed genes exhibited overexpression in HER2-positive tumors, consistent with prior studies documenting amplification not only of *ERBB2* but also of neighboring genes within the 17q12 locus, such as *GRB7* and *MIEN1* [33,34]. Interestingly, among HER2-low tumors, while overexpression of the *ERBB2* gene was observed compared to the HER2-negative group, this was not observed for the other two amplicon genes. This suggests that the *ERBB2* overexpression found in HER2-low cases is likely not due to genomic amplification, explaining their classification as HER2-negatives following the ISH test. Instead, it implies that *ERBB2* overexpression in these cases may be attributed to other molecular mechanisms linked to transcriptional regulation, resulting in their distinct dim to moderate IHC staining [28]. These mechanisms may involve epigenetic processes involving the acquisition of histone modifications, such as H3K4me3 and H3K9ac, which increases *ERBB2* transcription independently of gene amplification [35]. Additionally, it has also been shown that DNA hypomethylation of the HER2 gene body enhancer facilitates the binding of the transcription factor TFAP2C, thus enhancing *ERBB2* transcription [36]. In that sense, patients with tumors that have acquired epigenetic mechanisms that lead to higher *ERBB2* mRNA levels and a greater HER2 density in the cell membrane could potentially benefit from ADC-based therapy.

Moreover, in-depth molecular characterization analyses have revealed further differences between 1+ and 2+ HER2 groups [29]. It has been shown that 1+ tumors often exhibit a greater frequency of *TP53* mutations and an increased tumor mutational burden, resembling more of a basal phenotype, while 2+ tumors present a higher frequency of *ERBB2* mutations, along with higher *ERBB2* mRNA levels, resembling more of a HER2-enriched subtype [29]. Despite these reports, our results did not reveal differences in either clinicopathological features or mRNA expression of the HER2 amplicon genes between the HER2 1+ and 2+ subgroups. This indicates that the HER2-low subgroup evaluated in our study is fairly homogenous and can be analyzed and characterized together. Even so, it is possible that studies with larger cohorts might report higher levels of molecular heterogeneity among HER2-low tumors, making it difficult to establish clear clinicopathological distinctions from other HER2-expressing groups. This complexity explains why other studies have failed to report statistically significant differences in clinical features among these groups or why only subtle differences are usually found [37,38]. It is worth noting that this difficulty may also be attributed to other sources of variability, such as the study’s sample size and overall statistical power.

We acknowledge several limitations in this study, like the limited sample size, especially within the HER2 2+ group, potentially influencing the statistical analysis. Additionally, it is important to note that the current indication for ADC therapy is limited to advanced HER2-positive and HER2-low metastatic breast cancer [13,39], the latest subgroup with low representation in our study. However, we find the information gathered and presented in this study to be valuable. Our results highlight distinctions in disease outcomes based on different grades of HER2 IHC staining. Consistent with findings from larger-scale studies [38,40], we report more favorable outcomes for the HER2-low subset of breast cancer patients, possibly due to a higher frequency of ER-positive tumors among this group, although previous data suggest that the clinical benefit of ADC molecules for HER2-low patients is independent of ER status [13,39]. In that regard, we believe that identifying and characterizing this subset of breast cancer patients in the clinical setting may augment the scientific evidence regarding the potential of this group to benefit from novel therapies with ADC molecules.

## 5. Conclusions

Based on the new evidence regarding the important clinical benefits of ADC therapy within breast cancer patients with HER2-low expression, emerging evidence supports the subclassification and acknowledgment of this patient subset. Our investigation revealed that HER2-low breast cancer patients constitute a substantial proportion of diagnosed cases in the medical setting in our country. Additionally, these patients exhibit a distinct clinicopathological phenotype, resulting in better OS times compared to HER2-negative patients in the non-metastatic settings, which is probably related to a higher expression of the ER. Previous evidence highlights that this group constitutes a highly heterogeneous set of tumors, which is why further studies are still needed to focus on elucidating the molecular profiles of the various subgroups identified among HER2-low patients (HER2-low 1+ or HER2-low 2, whether ER+ or ER−). This will allow us to more clearly define the clinical implications of the HER2-low phenotype and will shed light on how the HER2-low group may continue to derive clinical benefits from new ADC molecules.

## Figures and Tables

**Figure 1 cancers-16-03141-f001:**
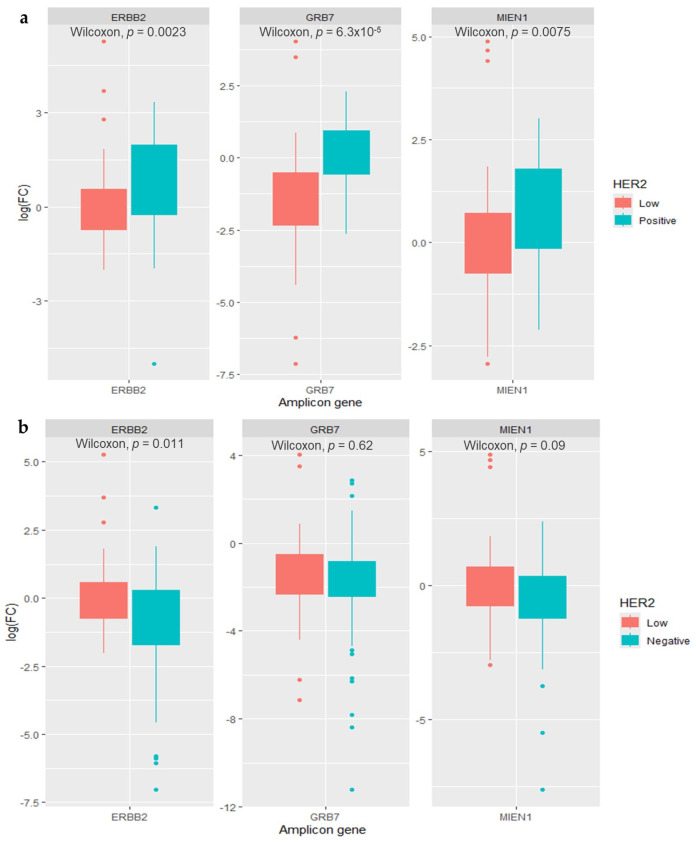
Comparison of mRNA expression levels of HER2 amplicon genes, *ERBB2*, *GRB7*, and *MIEN1*, between HER2-low tumors with (**a**) HER2-positive and (**b**) HER2-negative cases.

**Figure 2 cancers-16-03141-f002:**
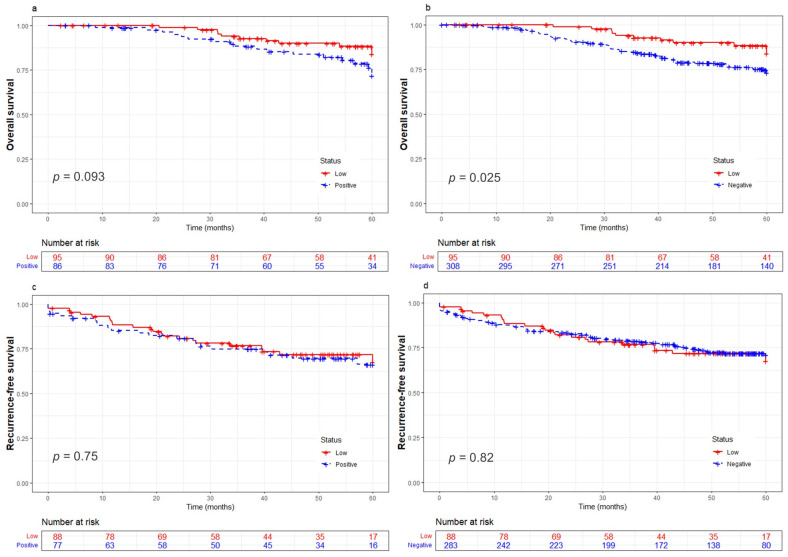
Differences in overall survival (**a**,**b**) and recurrence-free survival (**c**,**d**) between HER2-low tumors with HER2-positive (**a**,**c**) and HER2-negative (**b**,**d**) tumors.

**Table 1 cancers-16-03141-t001:** Clinicopathological characteristics of the study population.

	Level	Overall N (%)
n		516
Age of diagnosis	<50 years	140 (27.1)
	≥50 years	369 (71.5)
	Unknown	7 (1.4)
Breast cancer histological type	Invasive Ductal Carcinoma of No Special Type (NST)	516 (100)
AJCC clinical stage	I (I, Ia, Ib)	71 (13.8)
	II (IIa, IIb)	224 (43.4)
	III (IIIa, IIIb, IIIc)	199 (38.6)
	IV	13 (2.5)
	Unknown	9 (1.7)
Scarff–Bloom–Richardson	I	67 (13.2)
	II	263 (51.8)
	III	175 (34.4)
	Unknown	3 (0.6)
Tumor size	≤20 mm	140 (27.1)
	21–49 mm	191 (37.0)
	≥50 mm	137 (26.6)
	Unknown	48 (9.3)
Histological invasion	No	214 (41.5)
	Yes	263 (51.0)
	Unknown	39 (7.6)
Lymph node involvement	No	212 (45.5)
	Yes	254 (54.5)
Neoadjuvant treatment	Received	281 (54.4)
	Did not receive	225 (43.6)
	Unknown	10 (1.9)
Type of neoadjuvant therapy	Cytotoxic	186 (36.0)
	Hormonal	20 (3.8)
	Cytotoxic + Hormonal	33 (6.4)
	Cytotoxic + Trastuzumab	42 (8.1)
	Did not receive	225 (43.6)
	Unknown	10 (1.9)
Neoadjuvant treatment response	Complete	28 (5.4)
	Stable	38 (7.4)
	Partial	93 (18.0)
	Progression	43 (8.3)
	Unknown	79 (15.3)
Surgical management	Mastectomy	262 (50.7)
	Quadrantectomy	252 (48.8)
	Unknown	2 (0.4)
Type of adjuvant therapy	Cytotoxic	81 (15.7)
	Hormonal	204 (39.5)
	Cytotoxic + Hormonal	100 (19.4)
	Trastuzumab + Cytotoxic and/or Hormonal	65 (12.6)
	Did not receive	17 (3.3)
	Unknown	49 (9.5)
Radiotherapy	Received	424 (82.2)
	Did not receive	57 (11.0)
	Unknown	35 (6.8)
5-year clinical recurrence	No	330 (64.0)
	Yes	118 (22.9)
	Unknown	68 (13.2)
5-year vital state	Alive	389 (75.4)
	Deceased	100 (19.4)
	Unknown	27 (5.2)
Ki67 status	High (≥20%)	277 (53.7)
	Low (<20%)	239 (46.3)
HER2 status	Negative (0+)	325 (63.0)
	Low (1+/2+)	97 (18.8)
	Positive (3+)	94 (18.2)
Intrinsic subtype	ER+/HER2− (luminal A-like)	166 (32.2)
	ER+/HER2− (luminal B-like)	166 (32.2)
	ER+/HER2+	56 (10.9)
	ER−/HER2+	38 (7.4)
	ER−/HER2−	69 (13.4)
	Not classifiable	21 (4.1)

AJCC: American Joint Committee on Cancer. ER: estrogen receptor.

**Table 2 cancers-16-03141-t002:** Differences in clinicopathological characteristics between HER2-low tumors and HER2-negative and HER2-positive cases.

	HER2 Category	Low N (%)	Negative N (%)	*p* Value	Low N (%)	Positive N (%)	*p* Value
n		97	325		97	94	
Age of diagnosis	<50 years	20 (20.6)	83 (25.5)	0.392	20 (20.6)	37 (39.4)	0.008
	≥50 years	77 (79.4)	242 (74.5)		77 (79.4)	57 (60.6)	
AJCC clinical stage	I	15 (15.5)	47 (14.6)	0.977	15 (15.5)	9 (10.2)	0.390
	II	43 (44.3)	145 (45.0)		43 (44.3)	36 (40.9)	
	III/IV	39 (40.2)	130 (40.4)		39 (40.2)	43 (48.9)	
Scarff–Bloom–Richardson	I	13 (13.4)	43 (13.6)	0.612	13 (13.4)	11 (12.0)	0.015
	II	57 (58.8)	169 (53.5)		57 (58.8)	37 (40.2)	
	III	27 (27.8)	104 (32.9)		27 (27.8)	44 (47.8)	
Tumor size	≤20 mm	24 (24.7)	102 (31.4)	0.175	24 (24.7)	18 (19.1)	0.633
	21–49 mm	42 (43.3)	108 (33.2)		31 (32.0)	31 (33.0)	
	≥50 mm	31 (32.0)	115 (35.4)		42 (43.3)	45 (47.9)	
Histological invasion	No	40 (44.9)	132 (44.3)	1.000	40 (44.9)	42 (46.7)	0.935
	Yes	49 (55.1)	166 (55.7)		49 (55.1)	48 (53.3)	
Lymph node involvement	No	42 (44.2)	135 (46.9)	0.739	42 (44.2)	35 (42.2)	0.902
	Yes	53 (55.8)	153 (53.1)		53 (55.8)	48 (57.8)	
Neoadjuvant treatment	Received	55 (57.3)	174 (54.5)	0.721	55 (57.3)	52 (57.1)	1.000
	Did not receive	41 (42.7)	145 (45.5)		41 (42.7)	39 (42.9)	
Type of neoadjuvant therapy	Cytotoxic	39 (70.9)	126 (72.4)	0.053	39 (70.9)	21 (40.4)	<0.001
	Hormonal	4 (7.3)	15 (8.6)		4 (7.3)	1 (1.9)	
	Cytotoxic + Hormonal	5 (9.1)	27 (15.5)		5 (9.1)	1 (1.9)	
	Cytotoxic + Trastuzumab	7 (12.7)	6 (3.4)		7 (12.7)	29 (55.8)	
Neoadjuvant treatment response *	Complete	7 (17.9)	14 (11.1)	0.032	7 (17.9)	7 (18.9)	0.085
	Stable	5 (12.8)	30 (23.8)		5 (12.8)	3 (8.1)	
	Partial	24 (61.5)	53 (42.1)		24 (61.5)	16 (43.2)	
	Progression	3 (7.7)	29 (23.0)		3 (7.7)	11 (29.7)	
Surgical management	Mastectomy	49 (50.5)	157 (48.6)	0.831	49 (50.5)	56 (59.6)	0.266
	Quadrantectomy	48 (49.5)	166 (51.4)		48 (49.5)	38 (40.4)	
Type of adjuvant therapy	Cytotoxic	7 (7.8)	55 (19.5)	0.008	7 (7.8)	19 (24.4)	<0.001
	Hormonal	53 (58.9)	140 (49.6)		53 (58.9)	11 (14.1)	
	Cytotoxic + Hormonal	21 (23.3)	76 (27.0)		21 (23.3)	3 (3.8)	
	Trastuzumab + Cytotoxic and/or Hormonal	9 (10.0)	11 (3.9)		9 (10.0)	45 (57.7)	
Radiotherapy	Received	83 (92.2)	264 (86.3)	0.185	83 (92.2)	77 (90.6)	0.908
	Did not receive	7 (7.8)	42 (13.7)		7 (7.8)	8 (9.4)	
Ki67 status	High (≥20%)	48 (49.5)	153 (47.1)	0.764	48 (49.5)	76 (80.9)	<0.001
	Low (<20%)	49 (50.5)	172 (52.9)		49 (50.5)	18 (19.1)	
ER status	Negative	10 (10.3)	66 (20.3)	0.036	10 (10.3)	38 (40.4)	<0.001
	Positive	87 (89.7)	259 (79.7)		87 (89.7)	56 (59.6)	
PR status	Negative	18 (18.6)	94 (28.9)	0.058	18 (18.6)	49 (52.1)	<0.001
	Positive	79 (81.4)	231 (71.1)		79 (81.4)	45 (47.9)	
Intrinsic subtype	ER+/HER2− (luminal A-like)	26 (34.2)	140 (43.1)	<0.001	26 (34.2)	0 (0.0)	<0.001
	ER+/HER2− (luminal B-like)	47 (61.8)	119 (36.6)		47 (61.8)	0 (0.0)	
	ER−/HER2−	3 (3.9)	66 (20.3)		3 (3.9)	0 (0.0)	
	ER+/HER2+	0 (0.0)	0 (0.0)		0 (0.0)	56 (59.6)	
	ER−/HER2+	0 (0.0)	0 (0.0)		0 (0.0)	38 (40.4)	

AJCC: American Joint Committee on Cancer; ER: estrogen receptor; PR: progesterone receptor. * Patients with missing data or who did not receive neoadjuvant/adjuvant treatment or radiotherapy were not included in the statistical analysis.

**Table 3 cancers-16-03141-t003:** Risk of mortality of HER2-low tumors compared to HER2-positive and HER2-negative groups in a univariate and multivariate Cox-regression model.

Model	Univariate		Multivariate *	
HER2-low vs. HER2-positive				
Variable	HR (95% CI)	*p* value	HR (95% CI)	*p* value
HER2 status				
Positive	1.00	0.098	1.00	0.403
Low	0.53 (0.25–1.12)		0.69 (0.30–1.61)	
ER status				
Negative	1.00	0.0027	1.00	0.180
Positive	0.32 (0.15–0.67)		0.54 (0.22–1.32)	
Clinical stage				
I	1.00		1.00	
II	1.91 (0.23–15.9)	0.547	2.1 (0.25–17.7)	0.491
III/IV	6.57 (0.88–48.9)	0.065	5.7 (0.76–42.7)	0.089
HER2-low vs. HER2-negative				
Variable	HR (95% CI)	*p* value	HR (95% CI)	*p* value
HER2 status				
Negative	1.00	0.028	1.00	0.092
Low	0.49 (0.26–0.92)		0.57 (0.30–1.09)	
ER status				
Negative	1.00	<0.001	1.00	<0.001
Positive	0.23 (0.15–0.36)		0.26 (0.16–0.41)	
Clinical stage				
I	1.00		1.00	
II	1.88 (0.65–5.39)	0.238	1.84 (0.64–5.28)	0.255
III/IV	4.43 (1.6–12.2)	0.004	3.96 (1.43–10.9)	0.008

HR = hazard ratio; CI: confidence interval; ER: estrogen receptor. * The multivariate model was adjusted for the variables included in the univariate analysis.

## Data Availability

The original contributions presented in the study are included in the article/Appendix A, further inquiries can be directed to the corresponding author.

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
