# Peer review of "Characterization of HER2-Low Breast Tumors among a Cohort of Colombian Women"

_cancers, 2024, doi:10.3390/cancers16183141_

Round 1

Reviewer 1 Report

Comments and Suggestions for Authors

The Authors examined the clinical and pathological features, survival rates, and expression of HER2-related genes in Colombian patients with HER2-low breast cancer, comparing them to HER2-negative and positive groups.

The paper is overall quite well-written, but it explores a problem (Her2-low breast cancer) that is somehow mainstream in this field. Only in the last year and a half have more than 350 papers with a similar subject been published.

The Authors interestingly also performed a molecular analysis of the expression of the ERBB2 gene. Nonetheless, the results, even if accurate, are widely well-known.

I would question the reason this study was conducted on archival material (cases older than 5 years), which could raise concerns about the adequacy of the material.

For the HER2 evaluation of 1+ cases to be reliable, at least two expert pathologists blinded should have conducted it. Her2-low cases are sometimes very tricky. I also suggest stratifying these cases according to the percentage of 1+ cells to explore differences in this relatively new subgroup.

No mention is made of the histological analysis. An explanation on how the grading system is composed would be useful for those readers who use different systems. The inclusion criteria were "invasive ductal carcinoma (cited only in "sample selection")," which has not been a WHO category since 2014. A review of the selected cases according to more recent criteria would be appreciated and would have constituted a substantial result.

St. Gallen's consensus of approximation to molecular subtypes was updated in 2017.

The paper lacks the precise characterization of the cases selected and analyzed. 

Reviewer 2 Report

Comments and Suggestions for Authors

The manuscript compared the difference of clinical and pathological features, survival rates, and expression of HER2-related genes among patients with HER2-low, HER2-negative and positive breast cancer. In general, this paper is of interest and valuable. The novelty and necessity of this study are guaranteed. The paper is well designed with an adequate methodology description. The results are convinced. The manuscript may be eventually accepted for publication after addressing the following questions.

1. In line 16, the author described that “higher expression of the ERBB2 gene compared to HER2-positive tumors”, but a different statement was found in the Line 32, “HER2-low tumors showed higher mRNA expression of the ERBB2 gene and longer overall survival rates, compared to HER2-negative cases”. Is that contradictory?

2. Whether there is a difference between IHC 1+ and IHC 2+ (FISH negative) in terms of proliferation index, expression of the ERBB2 gene and overall survival rates?

3. The overall survival was related to clinical stage. The authors showed that the HER2-low presented longer overall survival than the HER2-negative. The clinical stages of these patients should also be compared.

Reviewer 3 Report

Comments and Suggestions for Authors

There are several fundamental points, without an answer to which it is difficult to continue reviewing an undoubtedly interesting and original manuscript:

1. Table 1 shows the response to neoadjuvant treatment, but does not say what kind of treatment it was. It is necessary to provide data on the drugs and the number of courses, as well as the extent of surgery performed, whether there was radiation treatment and adjuvant therapy. Subgroups should be compared according to these criteria too.

2. Did patients with HER2 1+ and 2+ amplification receive the same therapy as HER2-negative patients? When survival rates are compared, are they compared with all other conditions being equal, including the treatment regimen? How would the numbers change if the treatment regimen was changed to HER2-positive?

3. Table 3 should show in full both univariate analysis (for all factors!!!) and multifactorial analysis.

Round 2

Reviewer 3 Report

Comments and Suggestions for Authors

I have no further comments on the manuscript.